# Factors Regulating microRNA Expression and Function in Multiple Myeloma

**DOI:** 10.3390/ncrna5010009

**Published:** 2019-01-16

**Authors:** Irena Misiewicz-Krzeminska, Patryk Krzeminski, Luis A. Corchete, Dalia Quwaider, Elizabeta A. Rojas, Ana Belén Herrero, Norma C. Gutiérrez

**Affiliations:** 1The Institute for Biomedical Research (IBSAL), 37007 Salamanca, Spain; irenamk@usal.es (I.M.-K.); patrykk@usal.es (P.K.); lacorsan@usal.es (L.A.C.); dalia@usal.es (D.Q.); elirr@usal.es (E.A.R.); anah@usal.es (A.B.H.); 2Cancer Research Center-IBMCC (USAL-CSIC), 37007 Salamanca, Spain; 3National Medicines Institute, 00725 Warsaw, Poland; 4Faculty of Medicine, University of Salamanca, 37007 Salamanca, Spain; 5Hematology Department, University Hospital of Salamanca, 37007 Salamanca, Spain; 6Centro de Investigación Biomédica en Red de Cáncer (CIBERONC) number CB16/12/00233, 37007 Salamanca, Spain

**Keywords:** miRNA, myeloma multiple, miRNA regulation, methylation

## Abstract

Intensive research has been undertaken during the last decade to identify the implication of microRNAs (miRNAs) in the pathogenesis of multiple myeloma (MM). The expression profiling of miRNAs in MM has provided relevant information, demonstrating different patterns of miRNA expression depending on the genetic abnormalities of MM and a key role of some miRNAs regulating critical genes associated with MM pathogenesis. However, the underlying causes of abnormal expression of miRNAs in myeloma cells remain mainly elusive. The final expression of the mature miRNAs is subject to multiple regulation mechanisms, such as copy number alterations, CpG methylation or transcription factors, together with impairment in miRNA biogenesis and differences in availability of the mRNA target sequence. In this review, we summarize the available knowledge about the factors involved in the regulation of miRNA expression and functionality in MM.

## 1. Introduction

Multiple myeloma (MM) is a neoplasm characterized by the accumulation of clonal plasma cells (PC) in bone marrow, which is generally associated with the production of a monoclonal immunoglobulin and lytic bone lesions. MM accounts for 1% of all cancers and approximately 10% of all hematologic malignances. Although the incorporation of new drugs with different mechanisms of action has dramatically changed the perspective of treatment with an improvement in the median overall survival rate, it remains an incurable disease [1].

Similar to other cancer types, MM is a cytogenetically heterogeneous disease. Initial studies of conventional cytogenetics and fluorescent in situ hybridization (FISH), and more advanced studies based on microarrays and new generation sequencing (NGS), have revealed the existence of different types of genetic alterations that can be categorized into translocations, copy number abnormalities (CNAs) and point mutations [2,3,4]. The majority of chromosomal translocations affect *IGH* locus at 14q32, which occur in approximately 50% of MMs. Five chromosomal partners account for most of the *IGH* translocations in MM, 11q13, 4p16, 16q23, 6p21, and 20q11. These translocations lead to the overexpression of cyclin D1, *FGFR3/MMSET*, c-MAF, cyclin D3, and MAF-B, respectively. The presence of t(4;14), t(14;16) or t(14;20) translocations have being associated with poor prognosis. Moreover, a significant proportion of translocations in MM involves *MYC* oncogene (about 15%). CNAs are present in almost all MM patients; among them, the most frequent are gains of the entire chromosomes 5, 7, 9, 11, 15, 19, 21 and gains of 1q arm, and loss of 1p, 13q and 17p. Several studies have shown that 1q gains and 1p deletions are associated with short survival. Deletion of 17p, which contains the TP53 locus, is present in 10%, and remains a strong prognostic factor, which has been associated with a negative effect on survival in different treatment contexts. Whole-genome and whole-exome-based sequencing strategies have shown that there are few recurrently mutated genes in myeloma. Mutations affecting the MAPK pathway, including those in *KRAS*, *NRAS* and *BRAF*, are detected in approximately 40% of patients. Other mutations, such as those involving *FAM46C*, *TP53*, *DIS3*, *PRDM1*, *EGR1*, *TRAF3*, *CCND1*, *ATM*, *IRF4*, and *FGFR3*, were present in less than 10% of cases [5].

However, as demonstrated in other tumors, most of the changes in the gene expression in MM are apparently not preceded by alterations in DNA in the form of mutations, chromosomal anomalies or even epigenetic modifications (DNA methylation). The fact that steps of the embryological development depend on changes in gene expression and mainly occur without alterations in DNA supports the role of post-transcriptional regulation as a key mechanism for modifying gene expression levels in the absence of DNA abnormalities. Post-transcriptional gene regulation by microRNAs (miRNAs) has been the most widely investigated, since they can control the activity of more than 30% of all protein-coding genes. MicroRNAs are small regulatory RNAs, ranging from 19 to 25 nt, which are encoded from the genomic DNA. Most human miRNA genes are located between protein-coding genes (intergenic miRNAs), while about one-third of them have been found inside protein-coding genes, called host genes [6]. Moreover, there are miRNA genes, both inter- and intra-genic, that are clustered. Generally, there are between two and three miRNA genes in a cluster, but larger clusters have also been identified, like the *miR-17-92* cluster composed of six members [7]. The expression of miRNAs is regulated by multiple factors and molecular mechanisms, from those affecting the DNA copy number, methylation of CpGs, transcription factors, and miRNA biogenesis, to those modifying the miRNA binding site’s availability in the mRNA sequence.

In humans, as well as in other mammalians, almost 2000 different miRNAs have been identified, according to miRBase [8]. Although different functions have been described for miRNAs, the most relevant is the downregulation of gene expression at the post-transcriptional level by targeting specific messenger RNAs (mRNAs), either for degradation when fully paired to the seed region binding site at the 3′ untranslated region (3′UTR) of the mRNA target, or for inhibition of translation through partial base-pairing to complementary sites. Conversely, miRNAs might upregulate translation by other diverse mechanisms (reviewed in [9]).

MiRNAs are involved in critical biological processes, including cellular growth and differentiation, and can contribute to cancer pathogenesis (reviewed in [10,11,12,13,14]). In this regard, miRNAs have also been shown to be deregulated in MM. miRNA expression is deregulated in MM cells when compared to normal plasma cells [15,16,17], and several studies have demonstrated their key role in MM pathogenesis [18,19,20,21,22,23,24]. Below, a comprehensive review of the main mechanisms regulating miRNA expression and function, with focus on data available in MM, will be provided.

## 2. Copy Number Abnormalities Affect Expression of miRNAs

Mutations in the DNA sequences encoding miRNA is a rather rare event [25,26]. Genes, including those encoding miRNAs, can also be deregulated by copy number alterations of their loci (CNA). In primary MM samples, integrative analyses have identified a gene dosage effect induced by copy number changes on the expression of many mature miRNAs (Figure 1). One of the most frequent CNAs in MM is 1q gain, observed in more than 50% of patients [27]. The overexpression of miRNAs located at 1q, such as *miR-1231*, *miR-205*, *miR-215* and *miR-488*, was correlated with the gain of this chromosomal region [17]. *miR-215* directly targets MDM2, a negative regulator of p53 protein, and IGF-1 and IGF-1R, which control mobility and invasive properties of MM cells [28]. A cluster of *miR-520a-5p*, *miR-518d-5p*, *miR-498* and *miR-520g*, located at 19q, has been shown to be upregulated in those MM patients with 19q gains [17]. These miRNAs target p21, involved in cell cycle regulation [29], and FOXO3, a transcription factor implicated in cell growth, proliferation, and development of cancer [30]. However, their contribution to the pathogenesis of MM is unknown. Regarding chromosomal losses, monosomy and deletions of chromosome 13 are detected in up to 50% of MM cases [2,3]. Some of the miRNAs located at this region are downregulated in primary MM samples with del(13q), like *miR-15a/16* cluster and members of the *miR-17-92* cluster (*miR-17*, *miR19a* and *miR-20a*) [15,31]. *miR-15a* and *miR-16* downregulation contributes to MM pathogenesis by promoting cells growth and neoangiogenesis in bone marrow [31]. A high level of the *miR-17-92* cluster is, on the other hand, associated with poor prognosis in patients with MM, and it has been speculated that other factors may counteract the chromosome 13 deletion effect leading to the overexpression of these miRNAs in MM, as discussed below [32]. Furthermore, *miR-22*, encoded at the 17p locus, was significantly less expressed in those patients with the 17p deletion. [33]. Recently, *miR-22* has been identified as a tumor-suppressing miRNA and its expression is decreased in a variety of human neoplasms [34,35,36,37]. Moreover, the downregulation of 14 miRNAs in the presence of loss of heterozygosity (LOH) has been detected, such as *let-7b* at 22q, *miR-662* at 16p or *miR-140-3p* at 16q [17]. A probable consequence in MM can arise from the fact that *let-7* miRNAs (a family of 9 mature *let-7*, encoded by 12 different genomic loci (reviewed in [38])) function as a tumor suppressor through regulation of key oncogenes, including MYC and RAS [39,40]. Low expression of *let-7* family members is associated with poor prognosis in several cancer types [41,42]. In human myeloma cell lines (HMCLs), 61% of miRNA gene loci were affected by CNAs, showing a predominance of gains versus losses [33].

## 3. microRNA Expression and Epigenetics

Epigenetic modifications have a strong impact on gene expression regulation. Genes with methylated CpG in the promoters are typically repressed, while genes with unmethylated promoters can be expressed or not depending on other mechanism of expression regulation (reviewed in [43]). In many tumors, changes of DNA methylation have been found to be correlated with disease stage and patient survival (reviewed in [44]). In MM, the global DNA methylation level was observed to decrease from monoclonal gammopathy of undetermined significance (MGUS) to MM, although without further changes between newly diagnosed and relapsed cases [45,46]. Moreover, methylation of DNA encoding particular miRNAs has also been correlated with cancer progression. Increased methylation of the promoter regions of *miR-34a* [47], *miR-152*, *miR-10b-5p* [48] and *miR-203* was observed in MM cells compared to normal plasma cells [49]. In this regard, hypermethylation of *miR-34b/c* and *miR-129-2* has been detected during the transformation from MGUS to MM, as well as in MM relapse/progression [50,51]. Moreover, *miR-375* and *miR-342-3p* hypermethylation was also found in relapsed/progressed MM [52,53].

Many studies have been focused on analyzing the effect of the function restoration of those miRNAs downregulated by methylation. Such experiments, either using hypomethylating agents such as decitabine, or transfection with the particular miRNA, have demonstrated that forced expression of *miR-10b-5p*, *miR-34b*, *miR-34c-3p*, *miR-29b*, *miR-152*, *miR-194b* and *miR-203* resulted in decreased proliferation and induction of apoptosis [28,48,49,50].

Conversely, some miRNAs can directly affect epigenetic machinery by modulating the expression of enzymes, such as DNA methyltransferases (DNMT1, DNMT3a and DNMT3b) or histone methyltransferases. These miRNAs are often considered as a family of epigenetic miRNAs, “epi-miRNAs” [54]. In MM, *DNMT1* has been shown to be targeted by *miR-152*, whereas *miR-126* and *miR-140-3p* decrease *DNMT1* expression in other diseases [55,56]. Hypomethylation treatment or *miR-152* transfection of MM cells resulted in the decreased expression of *DNMT1*, *E2F3* [48]. The role of *miR-126* and *miR-140-3p* has been investigated in the interplay between myeloma cells and microenvironment. Co-culturing WL2 MM cells with bone marrow stromal cells (BMSC) revealed increased expression of *miR-126* and *miR-140-3p*, which led to repression of *DNMT1* and upregulation of *RANKL* [57]. The effect of the microenvironment on the enzymes responsible for DNA methylation is further supported by the experiments demonstrating increased expression of two DNA methyltransferases, DNMT3A and DNMT3B, after co-culture of MM cells with BMSC [58]. Moreover, expression of *DNMT3A* and *DNMT3B* is downregulated by *miR-29b* [59]. Overexpression of *miR-29b* reduced general DNA methylation two-fold. Interestingly, expression of *miR-29b* was increased by downregulation of histone methyltransferase *EZH2* [60]. *miR-29b* also targets the anti-apoptotic *MCL-1* protein [61], *CDK6*, *PSME4* [62] and *HDAC4* [21], which further supports links among epigenetics modification, drug resistance and apoptosis. A summary of epigenetically regulated miRNA is presented in Table 1.

## 4. Transcriptional Regulation of miRNA Expression

Intragenic miRNAs, located within introns or exons of protein-coding genes, are co-transcribed by Pol II and their expression is generally regulated by the same mechanisms as those of the host genes [6,70]. Little evidence exists about the relationship between expression of miRNAs and their host genes in primary MM samples, but so far no significant correlation between the intronic miRNAs and their host transcript expression has been detected [15]. Moreover, analysis of microarray-based datasets in various cell types, including MM, revealed that the large majority of intragenic miRNAs are not co-expressed with their host genes, and in MM only 11% of them were significantly correlated [71]. In HMCLs however, several associations have been described and are summarized in Table 2.

The intergenic miRNAs are transcribed from their own promoter by Pol II or Pol III, and a small group of miRNAs can be transcribed without any exogenous promoter [6,70,74]. Several transcription factors (TFs) have been described to regulate the intergenic miRNAs expression in MM (Figure 2).

A number of studies indicate that the bone marrow microenvironment (BMM) induces growth-promoting effects by modulating the expression of miRNAs [75,76]. The first published work regarding the regulation of miRNA expression in MM showed that interleukin 6 (IL6), the cytokine produced by bone marrow mesenchymal stem cells, induced changes in miRNA expression enhancing survival and drug resistance of myeloma cells. One of the miRNAs induced by IL6 is *miR-21*, whose stimulation is mediated by an upstream enhancer, the regulatory site that contains STAT3 binding sites. STAT3-mediated induction of *miR-21* has a pro-survival effect on MM cells, since ectopic increase of *miR-21* in IL6-dependent MM cell lines abolished the dependency on this cytokine [73]. STAT3 has been previously shown to induce the expression of anti-apoptotic genes, like survivin, *Bcl-2* or *MCL1* [77,78]. Moreover, downregulation of IL6 by treatment with berberine (a natural alkaloid derived from a traditional Chinese herbal medicine) decreased *miR-21* transcription through STAT3 down-regulation, which further validate the prosurvival impact of *miR-21* in MM [73]. Additional evidence of *miR-21* regulation by STAT3 was provided using ibrutinib (BTK inhibitor), which inhibits *miR-21* transcription by disrupting NF-kB and STAT3 binding to the *miR-21* promoter [79]. Moreover, BMM is highly hypoxic, which leads in MM cells to an AKT-mediated decrease of *miR-199-5p* expression, and consequently to increased expression of *HIF-1α* [80], a transcription factor involved in hypoxia response [81].

C-Myc is a transcription factor with recognized oncogenic functions. In MM, c-Myc deregulation is one of the key events associated with disease progression (reviewed in [82]). In order to regulate the transcription of the genes, c-Myc binds to DNA at multiple regulatory sites, one being the so-called E-Box. This evolutionary conserved DNA site was detected in the host gene of *miR-155* [68], one of the miRNAs essential for the B-lineage lymphocyte [83,84]. In fact, it has been demonstrated that c-Myc, Mad1 and Max regulate the expression of *miR-155* in MM by binding to this E-Box site [68]. In addition to *miR-155*, c-Myc activation results in widespread reprogramming of the miRNA expression pattern of tumor cells [85]. miRNAs belonging to the *miR-17-92* cluster were found to be significantly up-regulated only in MM samples but not in MGUS or healthy PCs [16] and the expression of this cluster is regulated by c-Myc in MM [86]. In this sense, it has been described that c-Myc inhibits apoptosis of MM cells by activation of *miR-17-92* cluster, which leads to the downregulation of the proapoptotic protein Bim [86]. Other targets of that miRNA cluster are *PTEN* and *E2F1*, which support its tumor-promoting functions [87,88,89,90,91]. Recently, the *miR-17-92* cluster was targeted with gapmeR (a chimeric antisense oligonucleotide that contains a central block of deoxynucleotide monomers sufficiently long to induce RNase H cleavage) antisense oligonucleotides, MIR17PTi, to induce degradation of MIR17HG primary transcripts and consequently prevent biogenesis of these miRNAs. As a result, MIR17PTi induced cell growth impairment, apoptosis, senescence and sensitized MM cells resistant to dexamethasone, melphalan or bortezomib. Moreover, treatment with MIR17PTi induced death only in cells with active MYC, through the upregulation of proapoptotic protein BIM and tumor suppressors p21, *TP53*, *E2F1* and *PTEN* [92]. This finding demonstrates the myc-dependent synthetic lethality of this compound in MM.

On the other hand, c-Myc takes part in a multi-component regulatory complex which trans-repress several miRNAs in MM, including *miR-23b*, *miR-29b* or *miR-29a* [75,93,94]. The repression of *miR-23b* and *miR-29b* in MM leads to activation of Sp1, a transcription factor that regulates the expression of cell-cycle, differentiation, and apoptosis-related genes [93]. Many miRNAs and transcription factors form autoregulatory loops, in which they mutually regulate each other [95]. A *miR-29b*-Sp1 feedback regulatory loop was described in MM [96]. Moreover, the treatment of MM cells with bortezomib decreased the expression of Sp1 with the consequent increase of *miR-29b* level, indicating that this miRNA can be regulated not only by c-Myc but also by Sp1 [75].

*TP53* is a tumor suppressor that interacts with numerous pathways and is commonly mutated in various cancers. However, in MM *TP53* mutations are not so frequent and p53 seems rather to be inactivated by other multiple factors (reviewed in [97]). Recent studies have demonstrated that miRNAs interact with p53 and this network regulates the p53 level, and, in turn, p53 also regulates the transcription, expression and the maturation of a group of miRNAs [28,98,99]. In MM, an autoregulatory loop between the *miR-194-2-192* cluster and p53 has been discovered, in which p53 acts as a transcriptional activator of *pri-miR-194-2* by directly binding to its core promoter element, and those miRNAs directly target 3′UTR of *TP53* [28]. Moreover, BMM induces expression of *miR-125-5p* in MM, leading to decrease of p53 expression and consequently, decrease of the p53-regulated *miR-194-2-192* cluster [100].

The knowledge of transcriptional regulation of miRNAs expression will help us better understand not only the function of miRNAs but also the effect of deregulation of transcription factors in MM.

## 5. Impaired miRNA Processing may Deregulate miRNA Expression

The biologically active miRNAs are generated in a two-step sequential mechanism involving two RNase III nucleases, Drosha and Dicer. The downregulations of these key factors in the biogenesis of miRNA has been reported in many cancer types (reviewed in [101]). Processing of the hairpin precursor (pre-miRNA) through Dicer generates a miRNA duplex, consisting of a miRNA (5p strand) and miRNA* (3p strand). The miRNA machinery is orchestrated by two major multiprotein complexes, the Drosha complex in the nucleus, and the cytoplasmatic RNA-induced silencing complex (RISC), which contains the Argonaute (Ago) family proteins as a core component (Figure 3) (reviewed in [12,14,102,103,104,105,106,107]). One of these strands generated by Dicer, known as the guide arm, will be incorporated into the RISC complex while the other, known as the passenger strand, will be degraded. One of the two strands is preferentially selected as a guide, either 5p or 3p. While either of the miRNAs strands can be used equally, the selection of the strand is highly regulated and depends on cell/tissue type [71,108]. The relationship between expression of sister miRNAs was compared in MM and other pathologies, such as acute lymphoblastic leukemia and prostate cancer, and the ratio between expression values of sister miRNAs was able to distinguish MM from other tissues [71]. The two mature miRNAs (3p and 5p) have partially reversed complementary sequences and tend to have different targeting properties [109].

The application of deep sequencing technologies to short RNAs and the analyses of pre-miRNA processing have led to the discovery of a distinct class of small RNAs known as moRNAs (miRNA-offset RNAs) derived from the sequence flaking 5′- or 3′arm pre-miRNA [110]. MoRNAs found in human tissues are generally included in the miRNA hairpin precursor [111]. Although the biogenesis and function of moRNAs remains unclear, it has been hypothesized that moRNA biogenesis might be associated with pre-miRNA processing [112,113]. Some data indicate that moRNAs may repress target transcripts as miRNAs do [114]. On the other hand, nuclear moRNA enrichment found in metazoan cells might indicate their role in regulating epigenomic modifications and transcription [115]. Likewise, the function of moRNAs in MM remains to be elucidated.

The ubiquitously expressed ubiquitin E3 ligase protein, cereblon, has been identified as a direct target of antimyeloma immunomodulatory drugs, like lenalidomide [116,117]. Remarkably, the Ago2 protein was found to be a substrate for the cereblon protein. Upon treatment of MM cells with lenalidomide, the level of Ago2 protein decreased with the consequent downregulation of global miRNA expression level. In line with this, the knocking down of *AGO2* and *Dicer* significantly decreased growth and viability of myeloma cells [118]. Moreover, silencing of *AGO2* led to decrease of the total miRNA level [118,119]. Moreover, Ago2 is involved in promoting angiogenesis by miRNA-mediated regulation of pro- and anti-angiogenic signals [120,121]. Expression of *AGO2* mRNA, however, did not significantly associate with global miRNA expression in primary MM cells [118]. *Dicer* mRNA level in NPC was very similar to MGUS, a pre-malignant condition, and significantly higher than in smoldering myeloma (SMM) and MM [122]. Moreover, higher expression of *Dicer* was associated with improved progression-free survival in symptomatic myeloma patients [122]. *Dicer* silencing led to cell cycle inhibition in MM cell lines [118], while higher *Dicer* expression was not associated with cell-cycle activation of proliferation in primary MM cells [122]. These results suggest that the effect of *Dicer* downregulation might be different in cell lines compared to the pathophysiological effect in primary MM. It is noteworthy that the miRNA expression level in MM is also affected by mutual crosstalk between myeloma cells and BMM, leading to the deregulation of certain miRNAs in MM [76].

Overall, these studies support that idea that the altered expression of proteins involved in the biogenesis of miRNAs may affect the MM pathogenesis.

## 6. Impact of miRNA–mRNA Associations

It is estimated that miRNAs repress the translation of approximately 30–50% of all the coding genes [123]. The effect of miRNAs usually induces a negative correlation between their expression levels and their target genes. However, this is not always the case and sometimes the regulatory effect is only observed at translation, with a low effect on the mRNA transcript levels [124,125]. miRNA-mediated gene regulation has a broad impact on gene expression since a single miRNA can control many transcripts and, in turn, a single transcript can be regulated by many miRNAs [126]. All these factors mean that the accurate identification of the targets regulated by miRNAs remains a challenge. Traditionally, miRNA–mRNA associations have been predicted using bioinformatic algorithms based on the sequence conservation between the miRNA and the target gene, such as miRanda [127], PicTar [128] or TargetScan [129]. More sophisticated approaches include other parameters such as the binding free energy and the secondary structure of the 3′UTR of the target gene. These improvements were implemented in tools such as PITA [130] or rna22 [131]. Nevertheless, even though these methods can identify the regulatory potential of miRNAs, it is noteworthy that they cannot reach the miRNA-targeted mRNA pairs with biological significance, as it depends on the association between the expression levels of both molecules and the biological context [132]. In recent years, the use of high-throughput techniques, such as microarrays, has enabled the detection of potential associations between miRNA and mRNA levels. However, these approaches alone are still limited at the time of revealing the mechanisms involved in the transcriptional cascade and how they are regulated. Many bioinformatic tools have been developed to integrate all these data through the generation of co-expression networks, using methods based on correlation [133] or Bayesian inference [134,135] or “mixed regulatory circuits reconstruction” [136]. These networks are built from expression data at both miRNA and mRNA level in order to identify critical genes in the network structure and to determine new regulatory relationships. Genes and miRNA are represented as the nodes of the network and the edge that connects two nodes represents the relationship between them. The aim of these methods is to detect the critical elements in the network, identifying the essential nodes and edges in the network structure. In summary, methods based on miRNA and mRNA expression integration have the major advantage of considering the biological background of the studied samples, however, all the strategies mentioned above require further validation. The in silico approach for validation is to use and compare different algorithms to predict the miRNA–mRNA interactions [137,138], which is widely recommended before the target validation experiments [138]. Several experimental strategies have been performed to determine the truthfulness and the biological significance of the miRNA-targeted mRNA interaction [139,140,141]. The most commonly used strategies are the validation of the expression of the miRNA and its targeted mRNA through qRT-PCR or Northern blot, and the demonstration of the miRNA interaction with a target site using the luciferase reporter assay [141]. However, although the use of these approaches is much extended in the literature, both have their pros and cons. The main disadvantage of co-expression assays is that they cannot differentiate direct and indirect miRNA targets [140], so it is not possible to know whether a miRNA directly interacts with an mRNA, or whether this effect is mediated by another target molecule. In the case of the luciferase reporter assay, the main drawbacks are that they depend on the cloning region and the variability of the protocol [140]. Because of this, it is necessary to combine different validation strategies to effectively determine the miRNA–mRNA interactions [140].

The expression of the miRNA and its targeted mRNA has also been actively studied in MM [15,17,132,142,143]. In silico analysis of the relationship between the miRNA and mRNA expression profiling in the context of the different MM cytogenetic subgroups has shown interactions involving genes related to biological pathways that could play a relevant role in the development of MM [15]. In the t(4;14) subgroup, *miR-135b* and *miR-146a* targeted *PELI2* and *IRAK1* genes, which are involved in the IL1 signaling pathway. A batch of four genes (*GPD2*, *GLCCI1*, *FNDC3B* and *ASH2L*) previously reported in cancer development were found to be targeted by the *miR-1* in the t(14;16) subgroup. Concerning the 13q deletion, the potential regulation of the MAPK pathway by miRNAs was highlighted. A final comparison among all cytogenetic subgroups revealed that some miRNA-mRNA interactions involving the *CCND2* gene were commonly dysregulated in the t(4;14), t(14;16) and 13q deletion subgroups, suggesting the potential targeting of *CCND2* by miRNAs in MM. Interactions were considered true positive when they were predicted by at least three out of the four algorithms used for the integrative analysis: TargetScan, miRDB, miRanda and Pictar. A similar study was performed through an integrative analysis with a correlation network of functional interactions [17]. This study identified more than 23,000 regulatory relationships involving 628 miRNAs and 6435 genes, which were considered real interactions if they were within the top 3% of the anti-correlated miRNA-targeted mRNA pairs. The most consistent sub-network was the corresponding to the t(4;14), which grouped seven miRNAs and 289 targets. In this sub-network, three of the target genes (*CBFA2T2*, *PP1R16B* and *GOSR2*) were commonly regulated by five miRNAs.

## 7. Single Nucleotide Polymorphisms and 3′UTR Polyadenylation Affect miRNA Binding Sites

The function of miRNAs may also be affected by sequence variations in miRNA binding sites, such as mutations and single nucleotide polymorphisms (SNPs) in the 3′UTR. These genetic variations inside 3′UTRs may overlap with miRNA binding sites and impair the translation inhibition or degradation of the mRNAs, or create new miRNA binding sites [144]. Computational analysis has identified more than 400 SNPs located within predicted and experimentally verified miRNA-binding sites [25]. Moreover, in recent years several studies have shown associations between SNPs in miRNA-binding-sites (miRSNPs) and cancer (reviewed in [145]). The first evidence for miRSNPs associated with hematological malignancies arise from a study in MM, in which the prognostic impact of six miRSNPs, located either in miRNAs target genes or in miRNA biogenesis pathway proteins, was evaluated. Only two miRSNPs, rs3660 in the *KRT81* gene and rs11077 in the *XPO5* gene, which were experimentally validated using renilla/luciferase reporter assays, were associated with better prognosis in MM after autologous stem cell transplantation [146] (Figure 4, left side).

On the other hand, another study based on computational prediction using genome-wide analysis and meta-analysis found five SNPs associated with MM risk [147] (Figure 4, right side).

Cancer cells often express substantial amounts of mRNA isoforms with shorter 3′UTR regions that usually result from alternative cleavage and polyadenylation (APA) [148]. The loss of miRNA-mediated repression due to the alternative cleavage and APA enhances the generation of shorter mRNAs isoforms that exhibit higher stability and produce more proteins [148,149]. This mechanism seems to be involved in the overexpression of *CCND2* in multiple myeloma. In fact, the shortening of *CCND2* 3′UTR by alternative polyadenylation with the consequent loss of miRNA binding sites was demonstrated both in myeloma cell lines and primary myeloma samples [150].

## 8. Regulation of miRNA Expression by miRNA Sponges

miRNA activity can be affected not only by the aforementioned factors, but also by regulatory RNA species, the so-called competing endogenous RNA (ceRNAs) or miRNA ‘sponges’. Sponge RNAs contain binding sites for miRNAs, and hence compete with target mRNAs for miRNA binding (Figure 5). RNAs that may function as sponges include long noncoding RNAs (lncRNAs), RNAs encoded by pseudogenes and circular RNAs (reviewed in [151]). Furthermore, highly abundant mRNAs have also been described to serve as endogenous sponges, since they may sequester miRNAs and suppress their activity over other mRNA targets. In general, miRNAs are stable and destroy targeted mRNAs. However, research suggests that endogenous targets may also control miRNAs by target RNA-directed miRNA degradation (TDMD) [152]. Interestingly, it has been described that some miRNAs can induce the degradation of specific circular RNAs after binding, whereas others are resistant and therefore act as strong miRNA sponges [153]. The ability to act as a miRNA sponge is also increased by the presence of multiple tandem high-affinity binding sites to a microRNA. Thus, the circular RNA sponge for *miR-7* (ciRS-7) contains >60 conserved *miR-7* seed matches and, therefore, can bind densely to this miRNA [154].

The influence of miRNA sponges in the regulation of cancer-associated genes is nicely illustrated by *PTEN*, one of the most frequently disrupted tumor suppressors in cancer [155]. Several protein-coding ceRNAs have been identified that share miRNA target sites with *PTEN*. Moreover, *PTEN* levels are also regulated by a *PTEN* pseudogene, *PTENP1*, which acts as a ceRNA by providing additional miRNA target sites [156]. Deregulation of these *PTEN*–ceRNAs seems to be responsible for *PTEN* under-expression in some tumors [157]. Other ceRNAs that influence cancer-related factors have been described [156], such as LncRNA *CRNDE*, which is overexpressed in colorectal cancer and promotes cell proliferation and metastasis [158,159]. Overexpression of *CRNDE* has also been found in MM and has been related to tumor progression and poor survival [160]. The authors reported that *CRNDE* induced the proliferation and an antiapoptotic response in MM by sponging *miR-451*. The lncRNAs, *CCAT1* and *MALAT1*, which are also overexpressed in MM [161,162,163], have been shown to promote MM cell growth by functioning as ceRNA for *miR-181a-5p* and *miR-509-5p*, respectively [161,164]. The lowered availability of these miRNAs increased the abundance of the tumor-promoting factors *HOXA1* and *FOXP1*, respectively.

In analogy with naturally occurring miRNA suppression mechanisms, artificially custom-designed sponges can be exploited for inhibition of miRNA activity (Reviewed in [165]). In fact, knockdown of overexpressed oncogenic miRNAs with this technology can be a therapeutic strategy for cancer [166]. Thus, in MM it has been reported that inhibition of *miR-19* and *miR-155* activity with “traditional” or artificially-improved sponges increases *SOCS1* levels, leading to enhanced activity of tumor suppressor p53 and inhibition of the transformed phenotype [167]. The deepest knowledge of miRNA sponges will pave the way to manipulate miRNA functions in MM and other cancer cells.

## 9. Conclusions

A full understanding of the role of miRNAs in MM pathogenesis requires not only exploration of the different effects of the deregulation of mature miRNAs, but also unveiling of the factors that regulate their expression and ultimately determine their function. These factors encompass the genomic modifications of DNA sequences encoding miRNAs, including DNA methylation, which is one of the most significant mechanisms that regulate miRNA expression in MM, the transcriptional regulation of miRNAs and, finally, variations in the mRNA target sites. Moreover, other elements, such as competing endogenous RNA and alterations in the complex enzymatic machinery that generate the mature and active miRNAs, may affect their final expression.

The understanding of the complex mechanisms of regulation that determine the expression and functionality of the mature miRNAs in MM may open new avenues for investigation, particularly in the field of therapeutic interventions.

## Figures and Tables

**Figure 1 ncrna-05-00009-f001:**
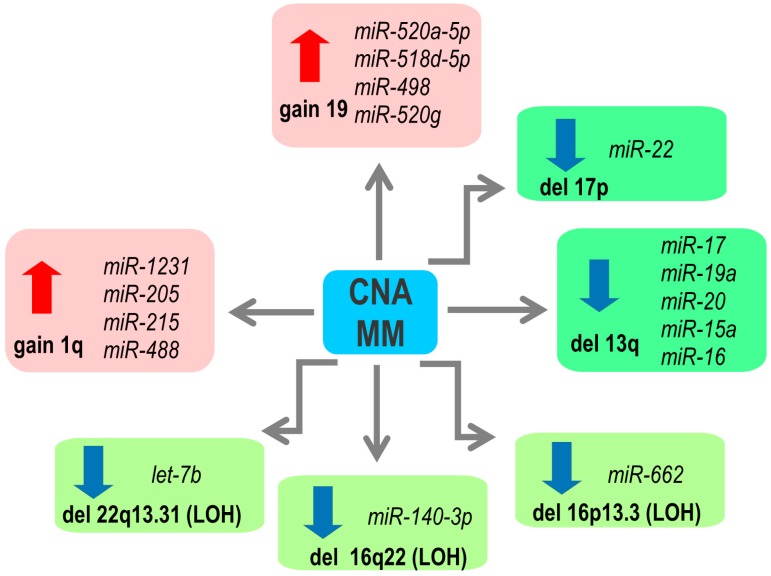
Copy number abnormalities (CNA) of genes encoding microRNAs (miRNAs) with impact on the miRNA expression in multiple myeloma (MM). LOH: loss of heterozygosity.

**Figure 2 ncrna-05-00009-f002:**
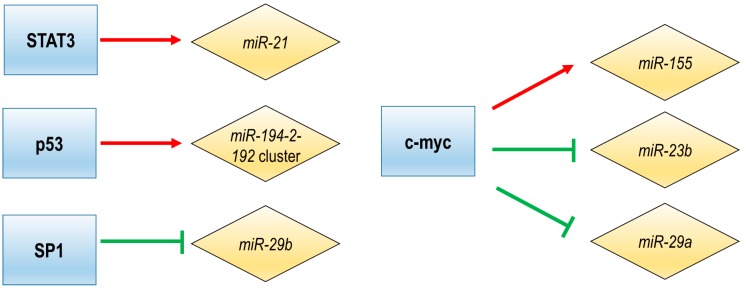
Regulation of miRNA expression by transcription factors (TFs) in MM.

**Figure 3 ncrna-05-00009-f003:**
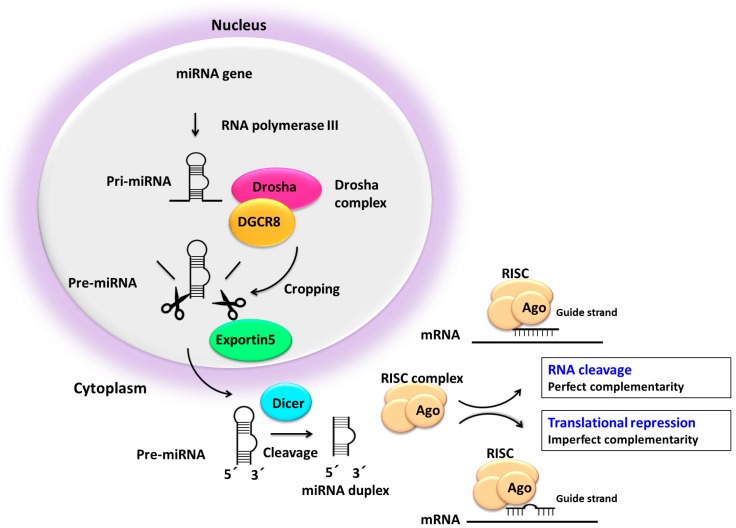
miRNA biogenesis pathway. RISC: RNA-induced silencing complex; Ago: Argonaute family proteins.

**Figure 4 ncrna-05-00009-f004:**
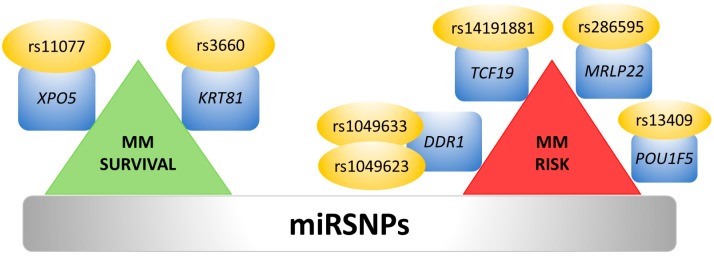
Relevant SNPs in miRNA-binding-sites (miRSNPs) studied in MM cells.

**Figure 5 ncrna-05-00009-f005:**
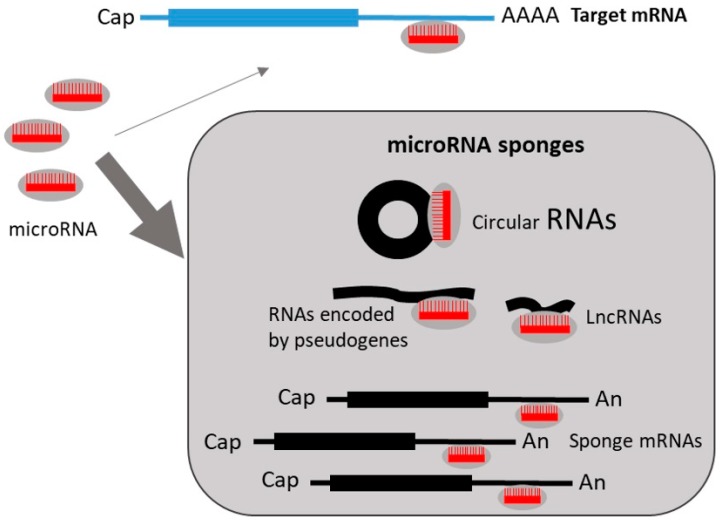
Regulation of miRNA activity on target mRNAs by miRNA sponges.

**Table 1 ncrna-05-00009-t001:** Brief summary of the DNA methylation status and function of miRNA in MM.

miRNA	Methylation Status in NPC, MM and HMCLs	Function in MM	Ref.
***9-3 and 9-1***	Unmethylated in MM, methylated in HMCLs	CD44 overexpression, (glycoprotein that has been associated with lenalidomide and dexamethasone resistance in myeloma)	[63,64]
***10b-5p***	Methylated in MM Upregulated after 5-aza-CdR in HMCLs	Not tested	[48]
***28***	Unmethylated in healthy controls and at MM relapse, methylated in some HMCLs	Inverse correlation between *CCND1* expression and *LPP/miR-28* methylation	[65]
***34a***	Methylated in primary MM, no difference between diagnosis and relapse, highly methylated in HMCLs	Inhibition of MM growth and reduction of bone lesions by targeting TGIF2	[47,66]
***34b/c***	Methylated in MM, increased methylation in relapsed MM, highly methylated in HMCLs	Inhibition of cellular proliferation and induction of apoptosis in MM cells	[50]
***124-1***	Methylated in primary MM, highly methylated in HMCLs	Repression of CDK6	[67]
***129***	Methylation increases from MGUS to MM at diagnosis and at relapse/progression	Downregulation of SOX4	[51]
***152***	Highly methylated in MM and in HMCLs Upregulated after 5-aza-CdR in HMCLs Targets *DNMT1*	Decrease of DNMT1 and E2F2 Induction of apoptosis in MM cells	[48]
***155***	Highly methylated in HMCLs Expression induced by LPS only when unmethylated in HMCLs	High expression of *miR-155* was associated with improved overall survival (OS) in MM.	[68]
***192, 194 and 215***	Methylated in MM cell lines	High expression of *miR-194* was associated with improved OS in MM. Overexpression associated with inhibition of cellular proliferation and migration	[28]
***198, 135a*, 200c, 125a-3p, 188-5p, 483-5p, 663, 630***	Methylated in MM Upregulated after 5-aza-CdR in HMCLs	Not tested	[69]
***203***	Methylated in HMCLs	Downregulation of CREB1 protein and inhibition of proliferation of myeloma cells	[49]
***342-3p***	Methylated in MGUS and increased methylation in relapsed MM Upregulated after 5-aza-CdR in HMCLs	Not tested	[52]
***375***	Methylated in MM and HMCLs	Downregulation of PDPK1 and IGF1R	[53]

Abbreviations: normal plasma cells (NPC), multiple myeloma (MM), Monoclonal gammopathy of undetermined significance (MGUS) and human myeloma cell lines (HMCLs).

**Table 2 ncrna-05-00009-t002:** miRNA co-regulated with their host genes and the mechanism described in MM.

Gene Locus	miRNA	Host Gene	Regulation Mechanism	Ref.
7q32.2	*miR-335*	*MEST*	Not by CNA	[33,72]
14q32.2	*miR-342-3p*	*EVL*	By promoter DNA methylation of its host gene, not by CNA	[33,72]
2q32.1-q32.2	*miR-561*	*GULP1*	Not by CNA	[33,72]
17q21.32	*miR-152*	*COPZ2*	Unknown	[33]
3q27.3-q28	*miR-28-5p*	*LPP*	By promoter DNA methylation of the host gene	[65]
17q23.1	*miR-21*	*TMEM49*	By an upstream enhancer, overlapping with intronic region of the host gene	[73]
21q21.3	*miR-155*	*BIC*	By promoter DNA methylation of the host gene	[68]

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
