# Peer review of "Factors Regulating microRNA Expression and Function in Multiple Myeloma"

_ncrna, 2019, doi:10.3390/ncrna5010009_

Round 1

Reviewer 1 Report

Comments

Overall, the content and organization of this review article have much room to be improved before further consideration for publication in the future.

1.       As it is a review about both miRNA and myeloma, a separate section of background information about myeloma is required. Therefore, it will be much easier for readers, in particular those who are not familiar with myeloma, to follow or to have a better picture when the discussion comes to those genetic alterations, such as t(4;14), t(14;16), t(11;14), del(17), and amp(1q21), etc.

2.       Regarding to miRNA expression in myeloma, a separate section on those articles about miRNA profiling in myeloma, and what did they found, should be highly relevant to how they were regulated.

3.       In contrast to the article title suggested that both expression and function would be discussed, most of the current content focused on expression only. More elaboration on the functional consequences of these deregulated miRNAs is desirable. For example, regarding to copy number variations, is there any literature demonstrating the function of these deregulated miRNAs in myeloma? This comment applies to each of the paragraphs in this article.

4.       Moreover, the implication arising from these data remains unclear. For example, this article has mentioned miRNAs that were regulated by transcription factors, could small molecule inhibitors targeting these transcription factors be a potential approach for myeloma therapy. As a piece of excellent review article, enough discussion on implications is essentially anticipated.

5.       Page 2 and 3, Section 2 and Figure 1 showed inconsistence on 1q gain, which should also be summarized in text, and LOH at 16q, which should also be shown in the figure.

6.       Regarding to regulation of intronic miRNAs, Page 5, Table 2, the authors may wish to provide further information on the specific regulation mechanism, such as CNV or methylation, for each of the intronic miRNAs. Moreover, regarding to LPP in Table 2, reference 48 was not about LLP/miR-28.

7.       In addition to some miRNAs could be transcribed by the associated transcription factors, and some were transcribed with their host genes, there are primary miRNAs, such as primary miR-34a, spanning a large genomic region. These should also be discussed.

8.       While each miRNA precursor may result in two mature miRNAs (5p and 3p), whether they share similar or different function should be discussed.

9.       Page 7, lines 176-182, the relationship between lenalidomide and the expression of AGO2 written here was unclear, please revise.

10.   Page 8, lines 190-193, DICER expression in MSCs seems out of scope. In contrast, is there any evidence of miRNAs in myeloma cells could be altered in the presence of MSC, and this should be a kind of regulation that worth to be discussed.

11.   While a lot of miRNA prediction algorithm have been discussed, currently used validation methods, and the pros and cons of each should also be included. In this connection, how were the miRNA-target pairs mentioned in lines 223-238 validated?

12.   Regarding to SNP on miRNA seed region binding site, page 9 lines 241-260, were these SNPs experimentally validated to affect binding of these miRNAs, and hence regulation of these genes? Otherwise, despite these SNPs were located at the miRNA seed region bind sites, this could be out of scope of this review.

13.   Regarding to “miRNA sponge”, a major unknown is the fate or downstream event of the corresponding miRNA. Any evidence on their fate upon binding with the “miRNA sponge”? Moreover, as obviously it is mediated by complementary base-pairing for both miRNA-target pair and miRNA-sponge pair, what is the mechanism leading to the presumed binding preference to the sponge instead of the target gene?

14.   Finally, a table summarizing miRNA function in myeloma is recommended.

Author Response

Reviewer 1

Overall, the content and organization of this review article have much room to be improved before further consideration for publication in the future.

1.      As it is a review about both miRNA and myeloma, a separate section of background information about myeloma is required. Therefore, it will be much easier for readers, in particular those who are not familiar with myeloma, to follow or to have a better picture when the discussion comes to those genetic alterations, such as t(4;14), t(14;16), t(11;14), del(17), and amp(1q21), etc.

According to the reviewer’s suggestion, we have included background information about MM in the introduction. Moreover, this section has also been modified according to other reviewer’s comments.  

2.      Regarding to miRNA expression in myeloma, a separate section on those articles about miRNA profiling in myeloma, and what did they found, should be highly relevant to how they were regulated.

Thank you for this comment, but we consider that “miRNA profiling in myeloma” is out of the scope of our review. In fact, this topic is comprehensively reviewed in the following papers:

- Pichiorri F, De Luca L, Aqeilan RI. MicroRNAs: New Players in Multiple Myeloma.

Front Genet. 2011 May 24;2:22

- Calvo KR, Landgren O, Roccaro AM, Ghobrial IM. Role of microRNAs from monoclonal gammopathy of undetermined significance to multiple myeloma. Semin Hematol. 2011 Jan;48(1):39-45

- Benetatos L, Vartholomatos G. Deregulated microRNAs in multiple myeloma. Cancer. 2012 Feb 15;118(4):878-87

- Lionetti M, Agnelli L, Lombardi L, Tassone P, Neri A. MicroRNAs in the pathobiology of multiple myeloma. Curr Cancer Drug Targets. 2012 Sep;12(7):823-37

- Dimopoulos K, Gimsing P, Grønbæk K. Aberrant microRNA expression in multiple myeloma. Eur J Haematol. 2013 Aug;91(2):95-105.

3.   In contrast to the article title suggested that both expression and function would be discussed, most of the current content focused on expression only. More elaboration on the functional consequences of these deregulated miRNAs is desirable. For example, regarding to copy number variations, is there any literature demonstrating the function of these deregulated miRNAs in myeloma? This comment applies to each of the paragraphs in this article.

We agree with the reviewer. We have added more information about the functional implications of the deregulated miRNAs in MM. The data has been added to “Copy number variations affect expression of miRNAs” (pages 3-4) and “DNA methylation and miRNAs expression” sections, including table 1.

4.      Moreover, the implication arising from these data remains unclear. For example, this article has mentioned miRNAs that were regulated by transcription factors, could small molecule inhibitors targeting these transcription factors be a potential approach for myeloma therapy. As a piece of excellent review article, enough discussion on implications is essentially anticipated.

Thank you for this comment. However, the effect of drugs or small molecules targeting transcription factors on miRNAs expression in MM was not considered when we defined the sections to be included in the paper. Therapeutic strategies based on miRNAs and its regulation could be a very interesting topic for other different review.

5.      Page 2 and 3, Section 2 and Figure 1 showed inconsistence on 1q gain, which should also be summarized in text, and LOH at 16q, which should also be shown in the figure.

The reviewer is right. We have corrected the figure and added missing information to this section. Moreover, we have slightly modified the presentation of the figure, according to other reviewer’s suggestion.

6.      Regarding to regulation of intronic miRNAs, Page 5, Table 2, the authors may wish to provide further information on the specific regulation mechanism, such as CNV or methylation, for each of the intronic miRNAs. Moreover, regarding to LPP in Table 2, reference 48 was not about LLP/miR-28.

Thank you very much for this comment. The mechanism of regulation of intergenic miRNAs, when available, has been added (table 2). The references have now been corrected.

7.      In addition to some miRNAs could be transcribed by the associated transcription factors, and some were transcribed with their host genes, there are primary miRNAs, such as primary miR-34a, spanning a large genomic region. These should also be discussed.

miR-34a is indeed a very important miRNA with potent tumor suppressor activity in many tumors. It has been poorly studied in MM from the point of view of regulation, except for the interesting approach of nanoparticles containing this miRNA. However, as commented before, therapeutic strategies based on miRNAs is out of scope of this review and there are experts in this field more qualified for reviewing this topic.

8.      While each miRNA precursor may result in two mature miRNAs (5p and 3p), whether they share similar or different function should be discussed.

According to the reviewer´s suggestion we have included in the manuscript a paragraph concerning 5p and 3p strands of miRNAs (page 10)

9.      Page 7, lines 176-182, the relationship between lenalidomide and the expression of AGO2 written here was unclear, please revise.

We have reorganized this paragraph according to the reviewer’s suggestion (page 11).

10.  Page 8, lines 190-193, DICER expression in MSCs seems out of scope. In contrast, is there any evidence of miRNAs in myeloma cells could be altered in the presence of MSC, and this should be a kind of regulation that worth to be discussed.

We agree. We have mentioned in the manuscript a brief sentence indicating the alteration of miRNAs in myeloma cells in the presence of bone marrow microenvironment (page 12, 293-295). Dicer expression in MSC has been eliminated.

11.  While a lot of miRNA prediction algorithm have been discussed, currently used validation methods, and the pros and cons of each should also be included. In this connection, how were the miRNA-target pairs mentioned in lines 223-238 validated?

A general description of validation methods for miRNA-mRNA interactions has been added to the main text and the comparisons among these methods have been included (pages 12-13). Experimental validations of some of the miRNA-mRNA interactions included in this section have been performed by renilla/luciferase reporter assays.

12.  Regarding to SNP on miRNA seed region binding site, page 9 lines 241-260, were these SNPs experimentally validated to affect binding of these miRNAs, and hence regulation of these genes? Otherwise, despite these SNPs were located at the miRNA seed region bind sites, this could be out of scope of this review.

Yes, miRSNP s3660 and rs11077 have been validated using renilla/luciferase reporter assays by de Larrea et al. This information has been added (page 14, 370).

The paragraph related to the relationship between miRSNPs and MM risk has been shortened according to reviewer’s suggestion, since the results have not been experimentally validated (page 14, 372-373).  

13.  Regarding to “miRNA sponge”, a major unknown is the fate or downstream event of the corresponding miRNA. Any evidence on their fate upon binding with the “miRNA sponge”? Moreover, as obviously it is mediated by complementary base-pairing for both miRNA-target pair and miRNA-sponge pair, what is the mechanism leading to the presumed binding preference to the sponge instead of the target gene?

To answer these questions the information has been incorporated to the review (page 15).

14.  Finally, a table summarizing miRNA function in myeloma is recommended.

Sorry, but the number of dysregulated miRNAs in MM is enormous and very hard to summarize in a table. Moreover, many miRNAs may have numerous functions and sometimes, opposite functions depending on the cellular context, in case of experiments with cell lines. Nevertheless, we have made a great deal of effort to add information about the function of some of the miRNAs included in the manuscript, according to reviewer’s comments.

Reviewer 2 Report

This is a very nice review which provides a thorough analysis of the mechanisms regulating miRNA expression in multiple myeloma.

I just recommend to implement the manuscript with the following papers in order to underline the relevance of the bone marrow microenvironment (BMM) in the regulation of miRNA expression.

-          The ability of bone marrow stromal cells to regulate miR expression via alteration of methylation of target genes could be added to the manuscript (Amodio et al., Expert Opinion Therapeutic Targets 2017; L. Yuan, G. C. F. Chan, BBA 2014).

-          The importance of hypoxia in regulating miRNA such as miR-199 (Raimondi et al., Biomed Res Int 2016; Raimondi et al., Oncotarget  2015) in the BMM, as well as the relevance of BMM to affect miR-21 expression (Pitari MR et al., Oncotarget  2015) or miR-125a-5p (Leotta et al., J Cell Physiol 2014) should be reported.

-          Regarding c-myc regulated miRNAs, the authours should mention the recent paper by Morelli et al., Blood 2018, which highlights  the capability of the miR-17-92 cluster to regulate a myc-based homeostatic loop.

-          Regarding the miRNA-mRNA association, the identification of miRNA-Transcription factors circuits in specific high risk patients would indeed add importance to the manuscript (Calura E et al., Oncotarget 2016);

-          Regarding the last paragraph, in which the authors discuss about lncRNAs, it is recommended that recent and relevant literature about lncRNA  and MALAT1 dysregulation in MM is cited (Nobili L et al., Oncotarget 2017; Amodio et al., Leukemia 2018; Hu Y, Lin J, et al., Leukemia 2018).

Author Response

Reviewer 2

This is a very nice review which provides a thorough analysis of the mechanisms regulating miRNA expression in multiple myeloma.

I just recommend to implement the manuscript with the following papers in order to underline the relevance of the bone marrow microenvironment (BMM) in the regulation of miRNA expression.

- The ability of bone marrow stromal cells to regulate miR expression via alteration of methylation of target genes could be added to the manuscript (Amodio et al., Expert Opinion Therapeutic Targets 2017; L. Yuan, G. C. F. Chan, BBA 2014).

Thank you for this suggestion. The microenvironment plays a pivotal role in the biology of many tumors, including MM. The effect of the microenvironment on DNA methylation in MM cells has been briefly discussed in the appropriate paragraph (page 5). Suggested references have been added.

- The importance of hypoxia in regulating miRNA such as miR-199 (Raimondi et al., Biomed Res Int 2016; Raimondi et al., Oncotarget 2015) in the BMM, as well as the relevance of BMM to affect miR-21 expression (Pitari MR et al., Oncotarget 2015) or miR-125a-5p (Leotta et al., J Cell Physiol 2014) should be reported.

We agree with the reviewer. The recommended literature is now discussed and cited (page 8)

- Regarding c-myc regulated miRNAs, the authours should mention the recent paper by Morelli et al., Blood 2018, which highlights the capability of the miR-17-92 cluster to regulate a myc-based homeostatic loop.

Thank you for the excellent suggestion.

We have added the information to the manuscript (page 9).

- Regarding the miRNA-mRNA association, the identification of miRNA-Transcription factors circuits in specific high risk patients would indeed add importance to the manuscript (Calura E et al., Oncotarget 2016);

We have included the “mixed regulatory circuits reconstruction” (MAGIA2) approach among the integration methods (page 12). The reference from Calura et al (2016) has also been included among the multiple myeloma miRNA-mRNA interaction studies (ref. 137).

- Regarding the last paragraph, in which the authors discuss about lncRNAs, it is recommended that recent and relevant literature about lncRNA and MALAT1 dysregulation in MM is cited (Nobili L et al., Oncotarget 2017; Amodio et al., Leukemia 2018; Hu Y, Lin J, et al., Leukemia 2018).

We agree with the reviewer. The recommended literature is now cited (ref. 154-155).

Reviewer 3 Report

The manuscript “Factors regulating microRNA expression and function in multiple myeloma” presents the factors impacting on miRNA expression and function, with focus on data available in MM.

The data collected are fairly complete, whereas the text of the manuscript and the organization of ideas need to be improved to convey a more lucid and logical vision, increasing the manuscript relevance.

Suggested revisions:

1)    The Abstract if poor, very general, weakly descriptive of the content. The sentence “… mostly similar to those of protein encoding genes” is very superficial and misleading, not convincing the reader that the review is really worth reading.

2)    The Introduction is chaotic and must be well revised to introduce the key concept in a logical order. Particularly,

a.     different things are mixed:

   i.     miRNA regulation and functions

   ii.     general concepts with data specifically in MM

b.     I do not appreciate the expeditious description of the miRNA biogenesis, since I am persuaded that the regulatory mechanisms reside in (or are linked to) the complexity and in the details of mRNA biogenetic (alternative) paths and steps.

c.     The structure of miRNA genes (miRNA genic, clustered, intergenic) needs to be introduced early in the manuscript, before epigenetic and transcriptional control of miRNA expression

d.     The last sentence “A comprehensive review of the main mechanisms regulating miRNA expression and functionality will be accomplished in this manuscript.” can better be: Below, a comprehensive review of the main mechanisms regulating miRNA expression and function, with focus on data available in MM, will be provided”.

e.     The sentence “They regulate gene expression at the post-transcriptional level …” is far too superficial according to the available knowledge, and must to be revised.

3)     P2 r56 skip “also”

4)     Figure 1, separate LOH, f.i. with a different shade of blue. Does CNV include LOH? Or CNV (gain or loss) + LOH can be better?

5)     P2 r71 not expressed à repressed?

6)     “DNA encoding miRNAs” is colloquial: try and find a less loose phrasing (several sentences across the man.)

7)     P3 r86 the fact that miRNAs can act as epigenetic regulators cannot be introduced by “In addition” ad I am not sure that this part fits the paragraph title (what about a separate Box?).

8)     P4 r97 “Taking together … ” and following is a suboptimal and probably too early conclusion.

9)     P5 r108 One third part à English!

10)   P5 r114 and around. Please consider and cite data on MM in the very pertinent paper: Biasiolo et al. Impact of host genes and strand selection on miRNA and miRNA* expression. PLoS One. 2011;6(8):e23854.

11)   P5 r1127-119 citation for POLIII missing? “One of the most important mechanisms of intergenic miRNAs expression regulation is mediated by transcription factors (TF) (Figure 2). Tautological and poorly phrased; need to think of a better presentation of Figure 2.

12)   The miRNA biogenesis part is “standard” does not mention alternative processing (e.g. moRNAs, see f.i. Bortoluzzi Set al.  MicroRNA-offset RNAs (moRNAs): by-product spectators or functional players? Trends Mol Med. 2011 Sep;17(9):473-4., tRFs, …) and post-processing modification and editing of miRNAs.

13)   P8 r200, more complex circuits have been considered (see the MAGIA2 method) also specifically in MM (Calura et al. Disentangling the microRNA regulatory milieu in multiple myeloma: integrative genomics analysis outlines mixed miRNA-TF circuits and pathway-derived networks modulated in t(4;14) patients. Oncotarget. 2016 Jan 19;7(3):2367-78.

14)   P8 r210, MiRNA-mRNA pairs only have biological interest if there is an association between the expression levels of both molecules, and it depends on the biological context [84]. à Need rephrasing

15)    P10 r275 see and maybe cite a review focused on blood cells (Bonizzato et. al Blood Cancer J. 2016). Moreover, the fact that miRNA can induce degradation of specific circRNAs whereas others are resistant and thud can be a good sponge is missing.

Author Response

Reviewer 3

The manuscript “Factors regulating microRNA expression and function in multiple myeloma” presents the factors impacting on miRNA expression and function, with focus on data available in MM.

The data collected are fairly complete, whereas the text of the manuscript and the organization of ideas need to be improved to convey a more lucid and logical vision, increasing the manuscript relevance.

 Suggested revisions:

1)      The Abstract if poor, very general, weakly descriptive of the content. The sentence “… mostly similar to those of protein encoding genes” is very superficial and misleading, not convincing the reader that the review is really worth reading.

Thank you for the suggestion. We have modified the abstract.

2)    The Introduction is chaotic and must be well revised to introduce the key concept in a logical order. Particularly,

a.     different things are mixed:

   i.     miRNA regulation and functions

   ii.     general concepts with data specifically in MM

b.     I do not appreciate the expeditious description of the miRNA biogenesis, since I am persuaded that the regulatory mechanisms reside in (or are linked to) the complexity and in the details of mRNA biogenetic (alternative) paths and steps.

c.     The structure of miRNA genes (miRNA genic, clustered, intergenic) needs to be introduced early in the manuscript, before epigenetic and transcriptional control of miRNA expression

d.     The last sentence “A comprehensive review of the main mechanisms regulating miRNA expression and functionality will be accomplished in this manuscript.” can better be: Below, a comprehensive review of the main mechanisms regulating miRNA expression and function, with focus on data available in MM, will be provided”.

e.     The sentence “They regulate gene expression at the post-transcriptional level …” is far too superficial according to the available knowledge, and must to be revised.

Thank you very much for this comment. We have reorganized the introduction according to these suggestions, and information about multiple myeloma has also been included as suggested by other reviewer.

3)     P2 r56 skip “also”

Thank you.

4) Figure 1, separate LOH, f.i. with a different shade of blue. Does CNV include LOH? Or CNV (gain or loss) + LOH can be better?

The reviewer is right. We have unified nomenclature as copy number abnormalities (CNA), which include gains and losses. LOH is considered separately. We have also corrected the figure in this regard. Moreover, the figure has been modified according to other reviewer’s suggestion.

5)     P2 r71 not expressed à repressed?

We agree, thank you.

6)     “DNA encoding miRNAs” is colloquial: try and find a less loose phrasing (several sentences across the man.)

We agree, thank you. It has been changed (page 4).

7)     P3 r86 the fact that miRNAs can act as epigenetic regulators cannot be introduced by “In addition” ad I am not sure that this part fits the paragraph title (what about a separate Box?).

We agree. We have changed the title in this regard.

8)     P4 r97 “Taking together … ” and following is a suboptimal and probably too early conclusion.

We have eliminated these conclusions. 

9) P5 r108 One third part à English!

We apologize for the mistake. It has been corrected.

10) P5 r114 and around. Please consider and cite data on MM in the very pertinent paper: Biasiolo et al. Impact of host genes and strand selection on miRNA and miRNA* expression. PLoS One. 2011;6(8):e23854.

Thank you for the suggestion. This reference has been added (ref 70, pages 7 and 10)

11) - P5 r1127-119 citation for POLIII missing?

The reference has been added (ref 73).

-         “One of the most important mechanisms of intergenic miRNAs expression regulation is mediated by transcription factors (TF) (Figure 2). Tautological and poorly phrased;

-         need to think of a better presentation of Figure 2.

We have corrected the sentence and the figure 2 (pages 7-8).

12)   The miRNA biogenesis part is “standard” does not mention alternative processing (e.g. moRNAs, see f.i. Bortoluzzi Set al.  MicroRNA-offset RNAs (moRNAs): by-product spectators or functional players? Trends Mol Med. 2011 Sep;17(9):473-4., tRFs, …) and post-processing modification and editing of miRNAs.

We would like to thank the reviewer for these useful comments to improve the paper. A paragraph addressing moRNA topic has been added to the manuscript (page 10).

13)   P8 r200, more complex circuits have been considered (see the MAGIA2 method) also specifically in MM (Calura et al. Disentangling the microRNA regulatory milieu in multiple myeloma: integrative genomics analysis outlines mixed miRNA-TF circuits and pathway-derived networks modulated in t(4;14) patients. Oncotarget. 2016 Jan 19;7(3):2367-78.

We have included the “mixed regulatory circuits” (MAGIA2) approach among the integration methods (page 12). The reference from Calura et al (2016) has also been included among the multiple myeloma miRNA-mRNA interaction studies (ref 137).

14)   P8 r210, MiRNA-mRNA pairs only have biological interest if there is an association between the expression levels of both molecules, and it depends on the biological context [84]. à Need rephrasing

The sentence has been corrected.

15)    P10 r275 see and maybe cite a review focused on blood cells (Bonizzato et. al Blood Cancer J. 2016). Moreover, the fact that miRNA can induce degradation of specific circRNAs whereas others are resistant and thud can be a good sponge is missing.

We thank the reviewer for this suggestion. The mentioned reference and information have been added (page 15).

Round 2

Reviewer 1 Report

This revision has been improved by incorporation of 1) background information about myeloma; 2) functional consequence about some of the dysregulated miRNAs; 3) specific mechanism for the regulation of different intronic miRNAs; 4) revamp of the relationship between lenalidomide and AGO2 expression; 5) removal of section about DICER expression in MSC cells; 6) further discussion of validation of miRNA target genes; and 7) clarification about miRSNPs with and without experimentally-validated binding.

However, here lists the issues to be resolved:

1.Many of the content were referenced to other reviews only, but not the original publications.

2.The meaning of this sentence in the manuscript was unclear, line 69: “In fact, all the steps in the embryological development depend on changes in gene expression programs rather than alterations in DNA. In this regard, the post‐transcriptional dysregulation is increasingly recognized as a key mechanism for modifying gene expression levels in the absence of DNA abnormalities”.

3.Line 83, “hundreds of different miRNAs have been identified” sounds outdated or incorrect.

4.Line 86, more professional terms, such as “seed region” and “seed region binding site”, should be used. Please revise.

5.Figure 1, the meaning of the directions of these arrows was confusing. For example, del(16q) pointed to MM CNA, whereas other arrows were pointed away from MM CNA.

6.Line 140, instead of implicated at relapse/progression, miR-129-2 was implicated at diagnosis from MGUS.

7.Table 2, reference 52 is not about miR-28. Please revise.

8.Page 10, line 254, what is downregulated in many cancer types? Please revise this sentence.

9.It remains unclear that whether the two mature miRNAs (5p and 3p) of a same precursor share similar or different functions.

10.Line 271. Could the authors elaborate on the role of miRNA-offset RNA? Any example? What is the rationale of the hypothesis about their origin from primary miRNA processing?

11.Section 6. The term of “miRNA-mRNA” is confusing. Do you mean the “mRNA” is the target gene of the corresponding miRNA? Similarly, line 330, it is not precise to describe miRNA and its target gene “co-expression”, as they should be associated with an inversely correlation. Please revise.

12.Lines 308 – 312 contain two consecutive sentences of similar meaning. Please revise.

13.Regarding to “miRNA sponge”, the fate of the sponged miRNA remains not discussed.

14.Finally, line 48, “FISH” but not “HISF”.

Author Response

Response to reviewer’s comments

Reviewer 1_Round 2

This revision has been improved by incorporation of 1) background information about myeloma; 2) functional consequence about some of the dysregulated miRNAs; 3) specific mechanism for the regulation of different intronic miRNAs; 4) revamp of the relationship between lenalidomide and AGO2 expression; 5) removal of section about DICER expression in MSC cells; 6) further discussion of validation of miRNA target genes; and 7) clarification about miRSNPs with and without experimentally-validated binding.

We thank the reviewer for agreeing with these improvements.

However, here lists the issues to be resolved:

1.      Many of the content were referenced to other reviews only, but not the original publications.

We agree. Nevertheless, we decided to use some reviews in order to simplify the large number of references, particularly in those paragraphs with general information, and to provide readers with more broad access to miRNA information. In several references we have now added the information in the text that the reference is a review.

2.      The meaning of this sentence in the manuscript was unclear, line 69: “In fact, all the steps in the embryological development depend on changes in gene expression programs rather than alterations in DNA. In this regard, the posttranscriptional dysregulation is increasingly recognized as a key mechanism for modifying gene expression levels in the absence of DNA abnormalities”.

The sentence has been clarified. (Page 2, 69-72)

3.      Line 83, “hundreds of different miRNAs have been identified” sounds outdated or incorrect.

We have updated the information according to miRBase data.

4.      Line 86, more professional terms, such as “seed region” and “seed region binding site”, should be used. Please revise.

The sentence has been changed accordingly. (Page 3, line 86)

5.      Figure 1, the meaning of the directions of these arrows was confusing. For example, del(16q) pointed to MM CNA, whereas other arrows were pointed away from MM CNA.

Thank you very much for detecting this mistake. We have corrected the figure.

6.      Line 140, instead of implicated at relapse/progression, miR-129-2 was implicated at diagnosis from MGUS.

The sentence has been changed. (Page 5)

7.      Table 2, reference 52 is not about miR-28. Please revise.

Thank you, the reference 52 has been removed from this table.

8.      Page 10, line 254, what is downregulated in many cancer types? Please revise this sentence.

The sentence has been changed. (Page 10, line 253)

9.      It remains unclear that whether the two mature miRNAs (5p and 3p) of a same precursor share similar or different functions.

This point has been clarified (page10, lines 265—267, Ref 109)

10.  Line 271. Could the authors elaborate on the role of miRNA-offset RNA? Any example?

Despite the limited knowledge, we have tried to shed light on possible functions of miRNA-offset RNA, according to the published data.

(Page 10, Ref 114 and 115)

What is the rationale of the hypothesis about their origin from primary miRNA processing?

The hypothesis of moRs biogenesis during Drosha processing of the pri-miRNA transcript has been raised by Shi W et al, 2009. Using small RNA sequencing they detected previously uncharacterized small RNAs that they called “moRs”, which arised from sequences located adjacent to the predicted pre-miRNA stem-loop. By alignment of coincident 5’ and 3’ moRs sequences from numerous miRNA loci the authors concluded that moRs arised from RNAse III processing.

moRNAs were also found in human tissues and are generally included in the miRNA hairpin precursor, and in some cases the moRNA overlaps the miRNA position by a few nucleotides (Langenberger D et al,2009).

We have added more information about moRs and reference to these papers (Page 10, Ref 110 and 111).

11.  Section 6. The term of “miRNA-mRNA” is confusing. Do you mean the “mRNA” is the target gene of the corresponding miRNA? Similarly, line 330, it is not precise to describe miRNA and its target gene “co-expression”, as they should be associated with an inversely correlation. Please revise.

Thank you for the suggestion. The term miRNA-mRNA has been modified throughout the text to avoid confusion. The term co-expression has been removed in the text when it may be confusing. (Page 13)

12.  Lines 308 – 312 contain two consecutive sentences of similar meaning. Please revise.

The repeated sentence has been removed.

13. Regarding to “miRNA sponge”, the fate of the sponged miRNA remains not discussed.

The information has been added. (Page 15 lines 392-394, Ref 152)

14. Finally, line 48, “FISH” but not “HISF”.

It has been corrected, thank you.

Reviewer 3 Report

The manuscript has been well revised.

Author Response

The manuscript has been well revised.

We thank the reviewer for agreeing with the improvements.
